# Current ionising radiation doses in the Chernobyl Exclusion Zone do not directly impact on soil biological activity

**Nicholas A. Beresford**[1,2]☯*, **Michael D. Wood**[2]☯, **Sergey Gashchak**[3]☯, **Catherine L. Barnett**[1]☯

**1** UK Centre for Ecology & Hydrology, Lancaster Environment Centre, Bailrigg, Lancaster, United Kingdom, **2** School of Science, Engineering & Environment, University of Salford, Manchester, United Kingdom, **3** International Radioecology Laboratory, Chornobyl Center for Nuclear Safety, Radioactive Waste & Radioecology, Slavutych, Kyiv Region, Ukraine

☯ These authors contributed equally to this work.
* nab@ceh.ac.uk

**Data Availability Statement:** Data are available from the NERC Environmental Information Data Centre (doi: 10.5285/19babe1c-b3a3-488c-b4fe-ebb4ab9237d8).

## Abstract

Although soil organisms are essential for ecosystem function, the impacts of radiation on soil biological activity at highly contaminated sites has been relatively poorly studied. In April-May 2016, we conducted the first largescale deployment of bait lamina to estimate soil organism (largely soil invertebrate) feeding activity *in situ* at study plots in the Chernobyl Exclusion Zone (CEZ). Across our 53 study plots, estimated weighted absorbed dose rates to soil organisms ranged from 0.7 μGy h$^{-1}$ to 1753 μGy h$^{-1}$. There was no significant relationship between soil organism feeding activity and estimated weighted absorbed dose rate. Soil biological activity did show significant relationships with soil moisture content, bulk density (used as a proxy for soil organic matter) and pH. At plots in the Red Forest (an area of coniferous plantation where trees died because of high radiation exposure in 1986) soil biological activity was low compared to plots elsewhere in the CEZ. It is possible that the lower biological activity observed in the Red Forest is a residual consequence of what was in effect an acute high exposure to radiation in 1986.

## Introduction

Soil organisms are essential for ecosystem function, playing a vital role in processes such as organic matter decomposition, nutrient cycling (and availability to plants) and bioturbation [1]. The effect of various stressors, including pollutants, on soil biological activity has been the focus of much study [e.g. 2–8]. Although it has been recommended that soil fauna could be used as radiological biomonitors [9], the effects of ionising radiation on soil biological activity at radiologically contaminated field sites has been relatively poorly studied. This may, in part, be the consequence of soil organisms (invertebrate macrofauna, mesofauna and microorganisms) typically being thought to be relatively insensitive to radiation compared to other biota [10, 11]. Some studies have reported effects on soil fauna at sites with high levels of natural radionuclides, including uranium mines [12–15]. However, it is likely that chemical toxicity

**Funding:** The work described in this paper was conducted as part of the TREE project (www.ceh. ac.uk/TREE), which was funded under the RATE Programme (https://www2.bgs.ac.uk/rate/home. html) by the Natural Environment Research Council, the Environment Agency and Radioactive Waste Management Ltd. (grant codes: NE/ L000318/1 and NE/L000520/1). All authors received funding via this route. The funders had no role in study design, data collection and analysis, decision to publish, or preparation of the manuscript.

**Competing interests:** The authors have declared that no competing interests exist.

rather than radiation dose is the cause of effects observed at such sites. Three months after the 1986 Chernobyl nuclear power plant accident, a 30-fold reduction in forest litter organisms was recorded at a pine forest site 3 km to the south of the Chernobyl reactor (total absorbed dose of 29 Gy estimated from thermoluminescent dosimeters placed in soil) compared to a similar site 70 km south with an estimated dose about 40-fold lower [16, 17]. Within 2–2.5 years after the accident, the biomass of soil fauna at the two sites was similar [16], but Geras'kin et al. [17] suggests that soil species diversity continued to be affected (approximately 20% reduction) 10 years after the Chernobyl accident.

In the longer-term after the accident, there is a lack of agreement in reported effects of radiation on soil organisms in the Chernobyl Exclusion Zone (CEZ); the CEZ is the approximately 4800 $km^2$ area around the Chernobyl Nuclear Power Plant (ChNPP) that was abandoned following the accident [18]. Mousseau et al. [19] deployed leaf litter bags at sites (forests or abandoned farmland reverting to forest) in the CEZ from September 2007 for nine months. Ambient dose rates ranged from 0.09 to 240 $\mu Sv\,h^{-1}$ (as determined by hand-held dosimeter) and the authors reported a significant reduction (by up to 40%) in leaf litter decomposition with increasing radiation levels; accounting for potential confounding variables, such as pH and soil moisture, did not change their conclusion. The authors also stated that the lower leaf litter decomposition resulted in the accumulation of organic matter in areas with higher radiation levels. Conversely, Bonzom et al. [20] found no negative impact on litter decomposition measured using litter bags at deciduous and mixed forest sites with ambient dose rates ranging from 0.22 to 29 $\mu Gy\,h^{-1}$ (the authors estimated that this equated to a maximum absorbed dose rate for litter decomposers of 150 $\mu Gy\,h^{-1}$); litter bags were deployed in November 2011 for a total of 318 d. It should be acknowledged that the maximum ambient dose rate studied by Bonzom et al. is about an order of magnitude below that of Mousseau et al. However, the data of Bonzom et al. suggested an increase in litter mass loss with increasing dose rate at dose rates where Mousseau et al.'s 'linear dose-response relationship' (*sic*) would predict a reduction in decomposition. Commenting on the contrasting findings of the Mousseau et al. [19] study and their own study, Bonzom et al. make the observation that the decomposition rates at some of the most contaminated sites of Mousseau et al. are comparable to, or higher than, those previously observed for litter from similar tree species at uncontaminated sites (see references cited by Bonzom et al.). Other studies considering ecologically important soil organisms within the CEZ found little or no effect of radiation [21–24].

In this paper we evaluate soil biological activity at sites across an ambient dose rate gradient of 0.6 to 237 $\mu Sv\,h^{-1}$ during spring 2016 (estimated absorbed dose rates to soil organisms 0.7 to 1753 $\mu Gy\,h^{-1}$). We used bait lamina [25] to measure soil organism feeding activity *in situ*. This approach has been used extensively to assess the impact of various chemical pollutants on soil biological activity [e.g. 3, 12, 26–28]. Bait lamina have been recommended as an indicator of soil biodiversity by a European Union working group [29] and as a method for use in ecological risk assessment [30]. Subsequently, the International Organization for Standardization (ISO) published ISO 18311:2016 [31], which defines a robust method for using bait lamina for field-determination of anthropogenic impacts on soil organism feeding activity.

All underlying data from the study reported here have been made openly available in a dataset published alongside this paper [32], enabling independent evaluation and reanalysis.

## Materials and methods

### Study sites

Eighteen study sites were selected, all of which were located within the CEZ. Eight of the sites were located in the Red Forest, an area of about 4–6 $km^2$ to the west of the Chernobyl nuclear

power plant where coniferous trees died as a consequence of high radiation exposure in the immediate aftermath of the accident [33]. At the time of this study, the area had regenerated with deciduous trees (predominantly *Betula* spp. (birch)) and generally sparse understorey vegetation including grasses, sedges, ericaceous species and, at a few sites, young pine saplings (site description notes can be found in the full dataset associated with this paper [32]). The decayed trunks of pine trees killed by the accident were also still evident on the ground surface. Within the Red Forest it was possible to select study sites with an approximately twenty-fold variation in ambient dose rate; the ambient gamma dose rate ($\mu$Sv h$^{-1}$) on the soil surface was measured at each study plot using a beta-shielded ATOMEX AT6130 dosimeter (https://atomtex.com/en/radiation-monitors/at6130-at6130a-at6130d-radiation-monitors), the reading being taken once the meter had stabilised. The meter is calibrated annually and has a high precision (difference in measurements of <3% based on repeat readings at sites within the CEZ). Three sites were selected in mature mixed deciduous woodland to the south of the Red Forest (Fig 1) with ambient dose rates within the range of those in the Red Forest. At two of these sites there was evidence (i.e. dead mature pine trunks on the ground surface) suggesting that of the few pine trees previously present at these sites some may have been killed by radiation following the accident; ambient dose rates at these sites were higher than at some of our Red Forest sites and they were located within the boundaries of the area where the complete death of pine trees was observed in 1986. Two deciduous woodland sites were also selected to the west of the Red Forest (on the edge of the 'western trace' of the initial release from the 1986 accident) with a further two sites in an area of comparatively low deposition to the north-west of the Red Forest (Fig 1). The deciduous woodland sites to the south and north-west were visually different to the Red Forest, having a more substantial litter layer which was virtually absent at some of the Red Forest sites. The deciduous woodland sites to the west were more similar to the Red Forest with a general lack of a litter layer and sparse understorey vegetation (S1 and S2 Figs). Three coniferous woodland sites were also selected in the area of comparatively low deposition (Fig 1). Although wildfires are common within the CEZ [18], none of the study sites had been impacted by wildfires (note many of the Red Forest and nearby deciduous forest sites were impacted by a wildfire in July 2016 after the study reported here [18]). The locations of all study sites were recorded using a Garmin 64st Global Positioning System (GPS) with an accuracy of ±3 m.

Permission for research activities and sampling within the Chernobyl Exclusion Zone was granted to Chornobyl Center by the State Agency of Ukraine for the Exclusion Zone Management.

## Bait lamina

The bait lamina strips were obtained from 'terra protecta GmbH' (http://www.terra-protecta.de/en/bait_strips.html). These semi-rigid polyvinyl chloride (PVC) strips (1 mm x 6 mm x 155 mm) have sixteen 1.5 mm apertures located at 5 mm intervals which begin 10 mm from the pointed lower end (the upper 70 mm of the strip has no apertures) [25, 31]. The apertures were filled with bait (for soil organisms) comprising 70% cellulose powder, 25% finely ground wheat bran and 5% activated charcoal.

Field application of the bait lamina followed ISO 18311:2016. At each of the sites, three 1x1 m plots with similar vegetation cover were identified; at a given site the plots were within c. 30 m of each other. At one of the deciduous woodland sites to the west of the Red Forest it was only possible to have two plots. Sixteen bait lamina strips were inserted into the ground within each plot (using a 4x4 grid with bait lamina approximately evenly spaced) during the period 17$^{th}$-19$^{th}$ April 2016. To ensure that the bait lamina would not be damaged during insertion into the soil, a thin bladed knife of similar dimensions to the bait lamina strip was used to cut a channel into the soil. The strips were then inserted such that the top aperture was *c*. 0.5 cm

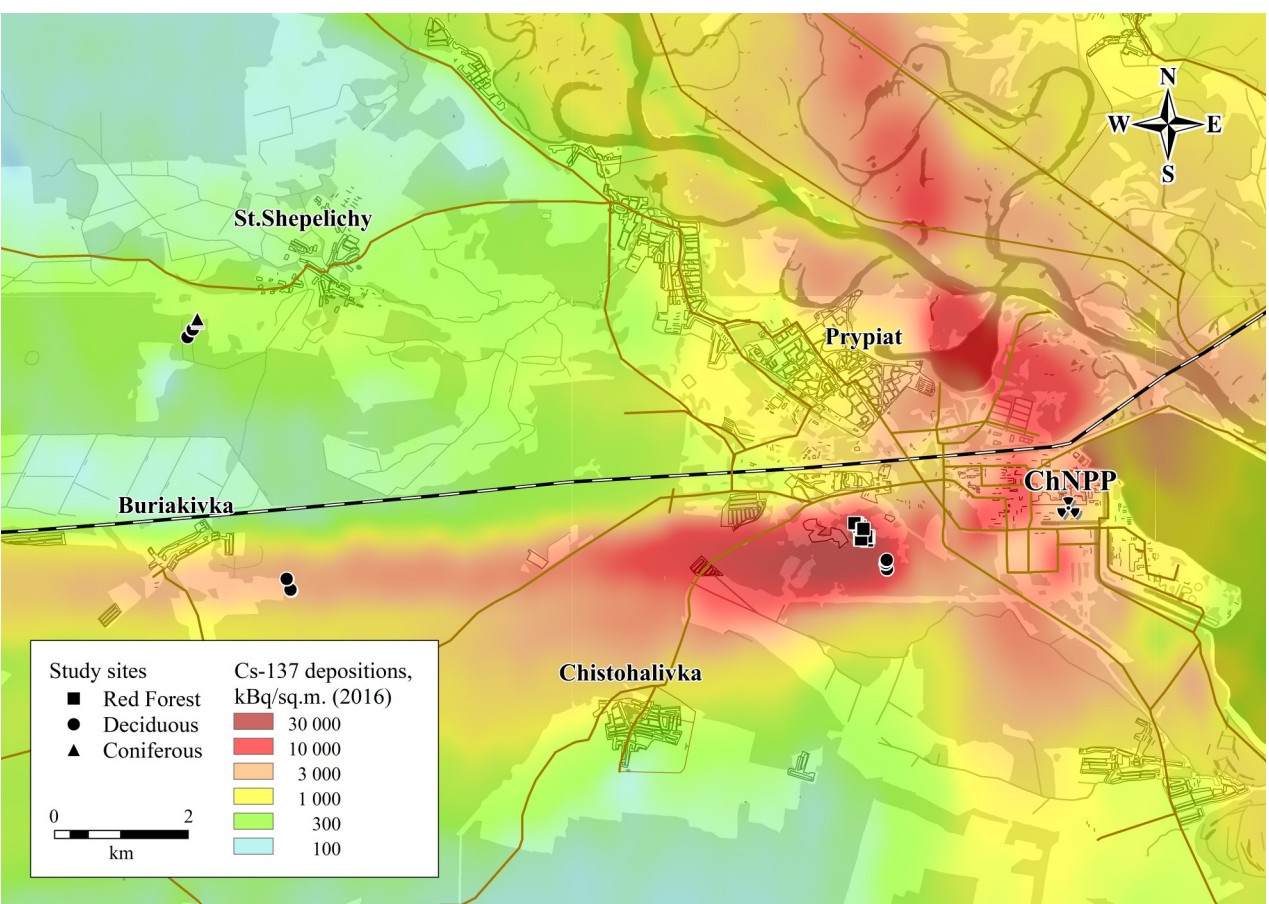

**Fig 1. Location of sample sites within the Chernobyl Exclusion Zone; deposition decay corrected to 2016.** Figure produced by and published with the permission of the Chornobyl Center.

below the soil surface. At each plot, an additional strip was inserted and withdrawn to determine whether soil abrasion would result in loss of bait from any of the 1.5 mm apertures. All bait lamina were retrieved after approximately 18 days (during 5th-7th May 2016). Soil temperature was measured in each plot at the time of bait lamina insertion and again at removal using a Eutech Instruments CyberScan pH300 (resolution 0.1˚C ± 0.3˚C).

After gentle wiping with tissue paper to remove adherent soil, the bait lamina sticks were visually assessed by placing them on an iPad tablet and using LightBox (v1.4) light table app. Feeding activity was assessed as qualitative feeding (i.e. the number of apertures showing any degree of bait consumption). A simple 'pierced' or 'unpierced' scoring system was used, where pierced was defined as clear evidence of bait removal (removal being distinguished from cracking). The vertical distribution of evidence of feeding was recorded. The bait lamina sticks were each read blind by two people (i.e. the people reading the strips had no knowledge of where they had been deployed within the CEZ). For analyses and discussion, results for an individual plot have been summed across all 16 bait lamina strips.

## Soil sampling and analyses

After removal of the bait lamina strips, five 10 cm deep soil cores (2.5 cm diameter) were collected from each plot (one from each corner and one from the middle). The five cores were

bulked, homogenised and sub-samples taken and sealed in zip-lock plastic bags for subsequent determination of pH and percentage moisture. The remaining soil sample was dried at 60˚C prior to radionuclide activity concentration determination. The pH of fresh soil was determined using the method of Allen [34] and percentage moisture was determined by mass loss when oven drying a sub-sample to constant mass at 60˚C. Five separate soil samples were also collected from each plot using a 10 cm deep collector of 250 cm$^3$ volume, which was driven into the ground until it was level with the surface; this gave a soil sample of known volume which was subsequently oven dried at 60˚C to enable the estimation of dry mass (DM) bulk density.

A Canberra-Packard gamma-spectrometer with a high-purity germanium (HPGe) detector (GC 3019) was used to determine the activity concentration of $^{137}$Cs. For calibration a standard $^{44}$Ti, $^{137}$Cs and $^{152}$Eu source comprising epoxy granules (<1.0 mm) with the density of 1 g cm$^{-3}$ was used (OISN-1; Applied Ecology Laboratory of Environmental Safety Centre, Odessa, Ukraine). The minimum detectable activities were 0.18 Bq $^{137}$Cs and 0.85 Bq $^{241}$Am per sample with uncertainties of around 10–15% and 20–30% respectively (p = 0.95); sample mass 5 g in petri dish geometry.

Americium-241 and $^{238,239,240}$Pu activity concentrations in soil samples (*c*. 5 g in Petri dish) were determined using a high purity germanium detector (Canberra GL0520R) with thin beryllium window (energy range of 5 to 700 keV) using the methodology described by Bondarkov et al. [35]. The $^{241}$Am activity concentration was estimated using the 59.5 keV gamma emission of its daughter isotope $^{237}$Np. The $^{238,239,240}$Pu activity concentration was estimated using measurement of the $L_x$-radiation (13–23 keV) emitted from excited uranium daughter isotopes following the α-decay of the Pu-isotopes. Because of absorption of low-energy emissions within the sample, a correction for self-absorption was used. The absorption correction was calculated assuming that the absorption ratio of the 13–23 keV U-isotope emission to the 661 keV emission of $^{137m}$Ba ($^{137}$Cs daughter) was similar to the absorption the ratio of the 32–37 keV $^{137m}$Ba emission to the 661 keV emissions of $^{137m}$Ba. For activity concentrations typically found at contaminated sites in the CEZ (>10 Bq g$^{-1}$) results obtained using this method have been shown to have good agreement with those from standard radiochemical methods (±10–15%) for both Pu-isotopes and $^{241}$Am [35]. To estimate radionuclide activity concentrations, sample spectra were compared to spectra for $^{238}$Pu, $^{239}$Pu, $^{240}$Pu and $^{241}$Am standards. The GL0520R detector was calibrated using a mixed gamma-standard (OISN-343 $^{137}$Cs, $^{152}$Eu and $^{241}$Am epoxy granules (<1.0 mm) with density of 1 g cm$^{-3}$; Applied Ecology Laboratory of Environmental Safety Centre, Odessa, Ukraine).

Strontium-90 activity concentrations in soil samples were measured spectrometrically without radiochemical pretreatment; for a detailed description of the method see Bondarkov et al. [36, 37] and Gaschak et al. [38]. The procedure used a β-spectrometer EXPRESS-01 (Nuclear Research of National Academy of Science, Ukraine) with a thin-film (0.1 mm) plastic scintillator detector calibrated using a $^{90}$Sr+$^{90}$Y standard (OISN-3 expoxy granules <1 mm, density of 1 g cm$^{-3}$; Applied Ecology Laboratory of Environmental Safety Centre, Odessa, Ukraine). Daily calibrations of the spectrometer were conducted; uncertainties were approximately 20% (2 sigma).

All equipment, methods and techniques used in the Chornobyl Center laboratory were officially certified and calibrated by State Enterprise 'KievOblDerzhStandard' (the state metrological service).

## Estimation of absorbed dose rates

We estimated absorbed weighted dose rates for the three relevant soil organisms which are available as defaults within the ERICA Tool version 1.2 [39, 40] and the revised 'R&D128'

spreadsheet model [41] available from https://wiki.ceh.ac.uk/display/rpemain/Ar+-+Kr+-+Xe +dose+calculator). The ERICA Tool default radiation weighting factors of 10 for alpha radiation, 3 for low energy beta and 1 for high energy beta and gamma radiation were applied [39]. The ERICA Tool was used to estimate doses to the 'Annelid' and 'Arthropod–detritivorous' reference organisms, whilst the 'R&D 128' spreadsheet model was used to estimate doses for soil bacteria (it is not possible to model bacteria within the ERICA Tool because of limitations on organism sizes). The two models have been shown to give reasonably consistent results for a given organism [42, 43]. To estimate internally incorporated radionuclide concentrations for annelids, and subsequently internal dose rates, concentration ratios as determined in 2014 for Lumbricidae species collected from a site at the western edge of the Red Forest were used [44]; the concentration ratio is the ratio between the fresh mass activity concentration of the whole body of an organism and the dry mass activity concentration of that radionuclide in soil. For detritivorous arthropods, default concentration ratios from the ERICA Tool were used; there was no need for concentration ratios for bacteria as all exposure is assumed to be external due to their small size. The full set of concentration ratios used in this study is presented in the accompanying dataset [32]. As total Pu activity concentrations in soil were reported, isotopic ratios from Red Forest soil samples collected in 2014 [44] were used to estimate $^{238}$Pu, $^{239}$Pu and $^{240}$Pu activity concentrations for inputting into the ERICA Tool. All three organisms were assumed to have a 100% occupancy within the soil column and hence a $4\pi$ exposure geometry. Measured soil dry mass percentages were used to correct for radiation attenuation within the soil matrix; percentage DM has a proportional influence on the estimated external dose rate, so a 10% soil DM would give an estimated external dose rate of 10% of that if soil were assumed to be 100% dry mass (see discussion in Stark et al. [45]).

## Statistical analyses

The Shapiro-Wilk test was used to test for normality of the data prior to subsequent statistical analyses. Tests included, paired t-tests, Kruskal-Wallis test, General Linear Model (GLM) fitting and regression analyses; all tests were performed using Minitab 18. The Red Forest is, in-effect, its own unique habitat, being an area where habitat destruction occurred in 1986. Although there has since been regeneration of deciduous tree species, at the time of this study the Red Forest was generally of poor habitat quality. We have therefore used three simplified habitat classifications for some of our data summarisation and subsequent analyses: 'Red Forest', 'deciduous' and 'coniferous'. Given that soil radionuclide activity concentrations at a given site varied by up to one order of magnitude, we have treated each plot as a separate observation point within our analyses (n = 53 plots) rather than averaging across the plots at a given site. Where it was necessary to transform feeding activity data to $\log_e$ values, feeding activity recorded as zero was assumed to be 0.1. For one GLM fitting it was necessary to use R v3.6.1 (see below).

## Results and discussion

### Soil radionuclide activity concentrations and dose rates

Table 1 summarises radionuclide activity concentrations by simplified habitat (coniferous, deciduous or Red Forest); data for individual sites and plots can be found in Barnett et al. [32]. Plutonium isotope activity concentrations in 17 of the 53 soil samples were below detection limits; for subsequent dose calculations, Pu-isotope activity concentrations that were below detection limits have been assumed to be the minimum detectable activity concentration.

   Estimated total absorbed dose rates for annelid, detritivorous arthropods and soil bacteria are presented by simple habitat type in Table 2 alongside measured ambient dose rate values.

**Table 1. Radionuclide activity concentrations in soils summarised by simple habitat.**

| | Simple habitat | Number of plots | Arithmetic mean | Arithmetic standard deviation | Minimum | Maximum |
|---|---|---|---|---|---|---|
| **Cs-137 Bq kg$^{-1}$ (DM)** | Coniferous | 9 | 5.76E+03 | 3.51E+03 | 2.88E+03 | 1.40E+04 |
| | Deciduous | 20 | 1.14E+05 | 1.30E+05 | 2.84E+03 | 4.22E+05 |
| | Red Forest | 24 | 4.12E+05 | 3.32E+05 | 2.93E+04 | 1.03E+06 |
| **Sr-90 Bq kg$^{-1}$ (DM)** | Coniferous | 9 | 1.83E+03 | 6.83E+02 | 9.00E+02 | 2.90E+03 |
| | Deciduous | 20 | 9.07E+04 | 1.73E+05 | 5.20E+02 | 7.83E+05 |
| | Red Forest | 24 | 1.68E+05 | 1.85E+05 | 1.15E+04 | 8.66E+05 |
| **Am-241 Bq kg$^{-1}$ (DM)** | Coniferous | 9 | 9.64E+01 | 2.69E+01 | 5.40E+01 | 1.26E+02 |
| | Deciduous | 20 | 5.76E+03 | 8.58E+03 | 6.30E+01 | 3.66E+04 |
| | Red Forest | 24 | 1.20E+04 | 9.73E+03 | 1.23E+03 | 3.24E+04 |
| **Pu-238 Bq kg$^{-1}$ (DM)** | Coniferous | 9 | 5.09E+00 | 2.44E+00 | <3.66E+0 | 1.14E+01 |
| | Deciduous | 20 | 5.60E+02 | 7.83E+02 | <2.00E+0 | 2.90E+03 |
| | Red Forest | 24 | 1.87E+03 | 2.24E+03 | 9.20E+01 | 9.49E+03 |
| **Pu-239 Bq kg$^{-1}$ (DM)** | Coniferous | 9 | 9.57E+00 | 4.59E+00 | <6.89E+0 | 2.14E+01 |
| | Deciduous | 20 | 1.05E+03 | 1.47E+03 | <4.00E+0 | 5.45E+03 |
| | Red Forest | 24 | 3.52E+03 | 4.21E+03 | 1.73E+02 | 1.79E+04 |
| **Pu-240 Bq kg$^{-1}$ (DM)** | Coniferous | 9 | 9.57E+00 | 4.59E+00 | <6.89E+0 | 2.14E+01 |
| | Deciduous | 20 | 1.05E+03 | 1.47E+03 | <4.00E+0 | 5.45E+03 |
| | Red Forest | 24 | 3.52E+03 | 4.21E+03 | 1.73E+02 | 1.79E+04 |

Internal exposure was estimated to contribute 21±8.5% (mean±SD) of the total annelid dose rate and 40±9.4% of the total arthropod dose rate; because of their small size internal exposure of bacteria is assumed to be negligible [41]. There was a significant relationship between ambient dose rate and the estimated total absorbed dose rates for all three organisms (p<0.001; $R^2$ = 74–89%). However, in all cases, the ambient dose rate was significantly lower that the total absorbed dose rates (p<0.001; paired t-test) (see Table 2); the differences between ambient dose rates and total absorbed dose rates were greatest for the soil bacteria and detritivorous arthropod (for which $^{137}$Cs, the dominant component of ambient dose, contributed less to the overall absorbed dose rate than for annelid). Therefore, whilst ambient dose rate is a good marker of comparative external exposure between sites, it would be erroneous to fit dose-response relationships based on ambient dose rate as others have often done for CEZ dose-

**Table 2. Measured ambient dose rate at the soil surface and estimated total weighted absorbed dose rate to selected relevant reference organisms summarised by simple habitat type.**

| | Simple habitat | Number of plots | Arithmetic mean | Arithmetic standard deviation | Minimum | Maximum |
|---|---|---|---|---|---|---|
| **Ambient dose rate (µSv h$^{-1}$)** | Coniferous | 9 | 6.09E-01 | 5.49E-02 | 5.00E-01 | 6.50E-01 |
| | Deciduous | 20 | 2.23E+01 | 2.63E+01 | 4.10E-01 | 7.80E+01 |
| | Red Forest | 24 | 1.01E+02 | 7.47E+01 | 1.23E+01 | 2.37E+02 |
| **Total absorbed dose rate (µGy h$^{-1}$) Annelid** | Coniferous | 9 | 1.57E+00 | 8.00E-01 | 8.40E-01 | 3.43E+00 |
| | Deciduous | 20 | 4.07E+01 | 5.00E+01 | 7.00E-01 | 1.84E+02 |
| | Red Forest | 24 | 1.45E+02 | 1.19E+02 | 1.04E+01 | 3.89E+02 |
| **Total absorbed dose rate (µGy h$^{-1}$) Arthropod** | Coniferous | 9 | 2.03E+00 | 9.12E-01 | 1.10E+00 | 4.09E+00 |
| | Deciduous | 20 | 6.15E+01 | 8.18E+01 | 1.00E+00 | 3.30E+02 |
| | Red Forest | 24 | 1.86E+02 | 1.50E+02 | 1.46E+01 | 4.71E+02 |
| **Total absorbed dose rate (µGy h$^{-1}$) Bacteria** | Coniferous | 9 | 4.28E+00 | 1.27E+00 | 2.51E+00 | 6.56E+00 |
| | Deciduous | 20 | 2.16E+02 | 3.30E+02 | 2.10E+00 | 1.41E+03 |
| | Red Forest | 24 | 6.03E+02 | 5.28E+02 | 4.90E+01 | 1.75E+03 |

effect studies [e.g. 19, 46–48]. For both the annelid and detritivorous arthropod, the largest contributor to absorbed dose was generally $^{137}$Cs. However, for detritivorous arthropod $^{241}$Am was estimated to contribute a similar percentage of the total dose rate as $^{137}$Cs in some cases. The differences between the dose estimated for annelid and detritivorous arthropod are largely due to the different concentration ratios used to determine organism activity concentrations and consequently the internal dose rate. For annelids we used values derived previously in the Red Forest [44] whereas, for detritivorous arthropod the ERICA Tool (version 1.3) default values [40] were used. With the exception of Pu, the default detritivorous invertebrate concentration ratios were higher than the Red Forest annelid values we have used. The choice of concentration ratio value is acknowledged to be a large contributor to uncertainty in estimated absorbed dose rates for wildlife [49, 50]. However, given the lack of concentration ratio data for detritivorous invertebrates in the CEZ, application of the ERICA Tool default values was necessary.

Unsurprisingly, mean absorbed dose rates (and soil activity concentrations) were, highest for the Red Forest (Tables 1 and 2). However, 13 of the 14 plots in deciduous woodland to the south and west of the Red Forest had estimated absorbed dose rates within the range of those estimated for the Red Forest.

The relative difference in estimated total absorbed dose rate for each of the three organisms was broadly consistent across plots. Any differences were due to variation in the isotopic ratios at a given plot. Therefore, in most of the following analyses we present and discuss dose rates for annelids only as conclusions are the same regardless of the organism; annelids have previously been shown to significantly contribute to the observed feeding activity on bait lamina strips [51].

## Bait lamina

Although there were some differences in the readings of the bait lamina between the two readers, these were insignificant ($p > 0.05$; paired t-test). Therefore, we have averaged the result of the two readers for use in statistical analyses (individual readings are presented in Barnett et al. [32]). Feeding activity was assessed as qualitative feeding (number of apertures showing any degree of bait consumption (bites)). Utilisation of the bait is summarised by simple habitat type in Table 3. The additional bait lamina strips used to test if inserting into the soil and withdrawing caused notable abrasion showed no damage to the bait.

**Table 3. Feeding activity summarised by simple habitat.**

| | Simple habitat | Number of plots | Arithmetic mean | Arithmetic standard deviation | Minimum | Maximum |
|---|---|---|---|---|---|---|
| **Apertures 1–16 (complete strip)** | Coniferous | 9 | 19.9 | 9.98 | 2.00 | 35.0 |
| | Deciduous | 20 | 20.8 | 15.6 | 2.00 | 53.0 |
| | Red Forest | 24 | 7.73 | 8.54 | 0 | 31.0 |
| **Apertures 1–8 (top of strip)** | Coniferous | 9 | 18.4 | 9.54 | 2.00 | 35.0 |
| | Deciduous | 20 | 15.7 | 11.6 | 2.00 | 45.5 |
| | Red Forest | 24 | 6.31 | 6.67 | 0 | 23.5 |
| **Apertures 9–16 (bottom of strip)** | Coniferous | 9 | 1.44 | 1.76 | 0 | 5.50 |
| | Deciduous | 20 | 5.10 | 5.39 | 0 | 16.5 |
| | Red Forest | 24 | 1.42 | 2.33 | 0 | 7.50 |

Feeding activity has been calculated as the total number of bites across the 16 bait lamina strips at each plot (i.e. a total of 256 apertures per plot). Results are presented for the complete strip (16 apertures per strip) and separately for the top and bottom eight apertures.

The feeding activity overall was relatively low compared to previous studies conducted elsewhere at sites of differing habitat types (deciduous woodlands, grasslands and arable land) across western Europe not impacted by pollutants [e.g. 26, 27, 52–54] with a mean of 6% of apertures showing evidence of feeding and a maximum for any plot of 21% (the total number of apertures per plot was 256). Four plots, all within the Red Forest, showed no evidence of feeding activity. The bait lamina are known to be an indication of soil faunal activity, with comparatively little contribution of microbial degradation to observed feeding activity [51, 55, 56]. Earthworms have been suggested as contributing significantly to the feeding activity observed on bait lamina sticks [51]. The soils across all of our sites are acidic (pH 3.9 to 4.9) and, in the case of the Red Forest sites and the deciduous woodland sites to the west, generally sandy in nature. Soil conditions such as these are known to result in low earthworm abundance [57] and consequently this is likely to contribute to the low overall utilisation rate of the bait lamina across all sites. The soil temperature at all plots was well above the lower bound of temperatures at which feeding on bait lamina has previously been reported [58].

In agreement with previous studies [e.g. 52–54, 59] utilisation of the bait was highest towards the soil surface (Table 3) with a significant difference in observed feeding activity between the top eight (*c*. 0.5 to 4 cm below soil surface) and bottom eight (*c*. 4.5 to 8 cm below soil surface) apertures (p<0.001; paired t-test). Given the localisation of soil organic matter in the uppermost layers of the soils in the CEZ (S1 and S2 Figs), the concentration of feeding activity in these upper soil layers is unsurprising.

### Estimated absorbed dose rate and feeding activity

There was a significant effect of habitat on feeding activity (p = 0.001; Kruskal-Wallis test). Median feeding activities in the Red Forest plots (3.5 bites) were approaching an order of magnitude below those of the deciduous (p<0.01; 20 bites) and coniferous (p<0.05; 22 bites) woodland plots (note that the bait lamina data were not normally distributed even when transformed to $\log_e$ (p>0.2), consequently the GLM fitting was performed using R assuming a Gamma distribution which allows for tails in the data). This difference in median feeding rate might be interpreted as suggesting an effect of radiation exposure on feeding activity. However, a simple linear regression of absorbed dose rate for all three organisms across all 53 plots showed no significant relationship with feeding activity (p>0.2; $R^2<0.03$) (Fig 2 presents

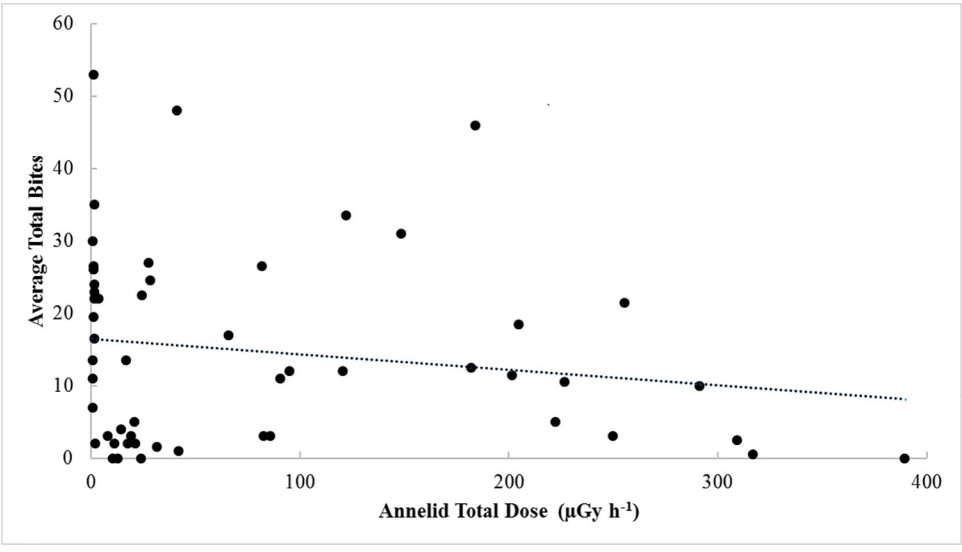

**Fig 2. A comparison of feeding activity (total bites) and estimated absorbed dose rate for annelid.**

annelid as an example). Repeating the regression using $\log_e$ transformed total absorbed dose rate and/or feeding activity data did not improve the significance. There was also no relationship if comparisons were restricted to plots within the same simple habitat type. Furthermore, feeding activity at the deciduous woodland sites to the south of the Red Forest, where estimated absorbed dose rates for annelids were in the range 28–180 µGy h$^{-1}$, were generally high (six of the nine plots being in the upper quartile of all observations) [32]. Conversely, at deciduous woodland sites to the west of the Red Forest, which had dose rates in the range 8–21 µGy h$^{-1}$, feeding activities were low with ≤5 bites in four of the five plots [32].

Our absorbed dose rate estimates are in-effect an average over the top 10 cm of the soil. However, radionuclide activity concentrations are highest in the upper soil layers [35, 60–62], so the concentration of feeding activity in the upper 4 cm of the soil profile suggests that we may be underestimating the total absorbed dose rates for the organisms with most feeding activity (see Beaugelin-Seiller [63] for discussion of heterogenous radionuclide distribution in soil/sediment profiles).

There are many environmental factors, which affect soil biological activity and which have previously been observed to influence feeding activity as determined using bait lamina strips. These include soil moisture, pH, organic matter content and soil temperature [12, 27, 59]. Whilst organic matter content was not determined for our study soils, bulk density was estimated. Harrison & Bocock [64] present a relationship between surface soil bulk density and organic matter content (the higher the soil bulk density the lower the organic matter content). Consequently, we can assume our estimated soil bulk densities are proxies for organic matter content. Soil pH, percentage moisture, bulk density and temperature values for the study plots are summarised in Table 4; there was no significant difference between April and May soil temperatures ($p > 0.3$; paired t-test) and consequently Table 4 presents values averaged across the two measurement times. All measured soil parameters (pH, percentage moisture, bulk density and temperature) show a significant effect of simple habitat ($p < 0.003$; Kruskal-Wallis test). Regressions of feeding activity against the soil parameters gave significant relationships for percent moisture ($p < 0.001$; $R^2 = 0.36$) and soil bulk density ($p < 0.001$; $R^2 = 0.40$) (Fig 3); the bulk density relationship in-effect implies that feeding activity increased with increasing soil organic matter content. Whilst the regression of feeding rate against pH was significant, the amount of variance this explained was poor ($p < 0.05$; $R^2 < 0.09$). There was no significant

**Table 4. Soil pH, percentage moisture, bulk density and temperature summarised by simple habitat.**

|  | Simple habitat | Number of plots | Arithmetic mean | Arithmetic standard deviation | Minimum | Maximum |
|---|---|---|---|---|---|---|
| **pH** | Coniferous | 9 | 3.87[a] | 0.10 | 3.74 | 4.05 |
|  | Deciduous | 20 | 4.33[a] | 0.52 | 3.52 | 5.08 |
|  | Red Forest | 24 | 4.58[b] | 0.25 | 3.94 | 4.93 |
| **% moisture** | Coniferous | 9 | 19.3[a] | 5.17 | 10.6 | 26.4 |
|  | Deciduous | 20 | 23.7[a] | 12.7 | 7.07 | 49.6 |
|  | Red Forest | 24 | 9.43[b] | 2.06 | 4.40 | 13.6 |
| **Soil bulk density (g cm$^{-3}$)** | Coniferous | 9 | 0.78[a] | 0.13 | 0.57 | 0.92 |
|  | Deciduous | 20 | 0.86[a] | 0.30 | 0.39 | 1.38 |
|  | Red Forest | 24 | 1.24[b] | 0.08 | 1.10 | 1.39 |
| **Soil temperature (°C)** | Coniferous | 9 | 9.52[a] | 0.26 | 9.05 | 9.90 |
|  | Deciduous | 20 | 9.68[a] | 0.81 | 8.15 | 11.5 |
|  | Red Forest | 24 | 10.8[b] | 1.57 | 9.35 | 15.5 |

For a given parameter significant differences ($p < 0.05$; generalised linear model) between habitats are identified by different superscripted letter (a,b).

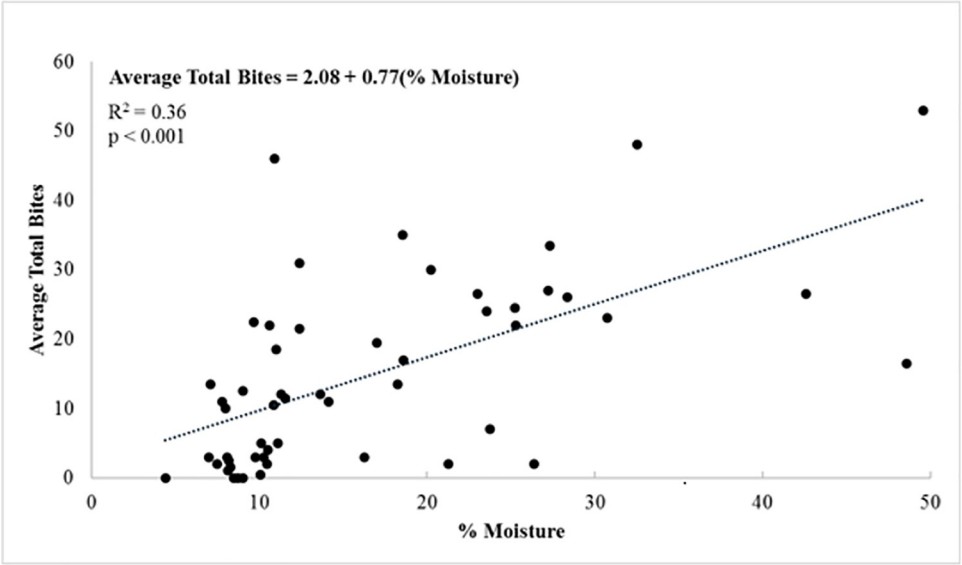

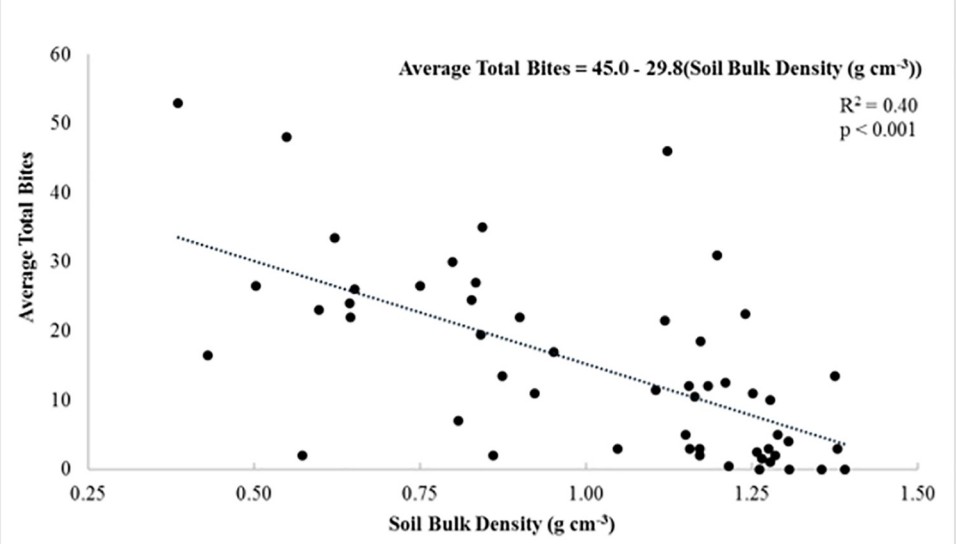

**Fig 3.** Relationship between feeding activity (total bites) and soil percentage moisture (top) and soil bulk density (bottom).

relationship between feeding rate and soil temperature ($p > 0.1$; $R^2 = 0.04$). Soil temperature may have been in part determined by the time of day when measurements were made; for both the April and May measurements there is a trend in the data suggesting that soil temperature increased with time of day ($p < 0.01$; $R^2 = 0.18–0.50$). However, there was no consistent bias in the time of day at which sites in different habitat classifications were visited. Consequently, we have not considered soil temperature further in our analyses. We note that there is a general lack of soil parameter data available for the CEZ and hence the data from the present study (see [32] and Table 4) makes a valuable contribution.

Given there was an effect of habitat/soil parameters on the feeding activity we observed, we have conducted regression analyses of feeding activity against absorbed dose rate with soil

**Table 5. Statistics of regression for annelid.**

|  | DF | Adjusted sum of squares | Adjusted mean squares | F-Value | P-Value |
|---|---|---|---|---|---|
| Regression | 45 | 6.32E+03 | 1.40E+02 | 0.33 | 0.988 |
| Total absorbed dose rate (μGy h$^{-1}$) Annelid | 1 | 2.62E+01 | 2.62E+01 | 0.06 | 0.810 |
| pH | 44 | 6.06E+03 | 1.38E+02 | 0.33 | 0.989 |
| Error | 7 | 2.94E+03 | 4.20E+02 |  |  |
| Total | 52 | 9.26E+03 |  |  |  |

moisture, soil pH and soil bulk density included as continuous predictors. The resultant statistics (presented for the example of annelid in Table 5) demonstrate no significant interaction between feeding rate and dose rate when these other variables were taken into account. A lack of relationship between feeding activity and absorbed dose rate was also found for the detritivorous arthropod and soil bacteria (p>0.2); by estimating the absorbed dose rates for the smallest (bacteria) and largest soil organisms (annelids) we should have encompassed the range of likely dose rates to any organism feeding on the bait lamina (e.g. nematodes, collembola, mites [65]). We repeated the analyses without the soil parameters, but with simple habitat as a categorical predictor. Again, there was no significant influence of absorbed dose rate on feeding activity (p>0.3 in the case of the annelid).

## Conclusions

Our study did not find any effect of current radiation exposure on soil biological activity as determined by bait lamina strips. This is in agreement with the leaf litter decomposition study of Bonzom et al. [20] but disagrees with the conclusion of Mousseau et al. [19] that '*we have shown severely depressed levels of litter mass loss in the most contaminated forest areas around Chernobyl*' (sic). A criticism of the Bonzom et al. study could be that it considered a lower dose rate range than the work of Mousseau et al. However, the maximum ambient dose rate across our sites was comparable to that quoted by Mousseau et al. Furthermore, from the map of study sites presented in Mousseau et al. and their supplementary data table, it would appear that a number of their sites were in similar locations to those used in our study.

The study presented here is the largest deployment of bait lamina reported to date within the CEZ. To our knowledge bait lamina have been used in the CEZ twice before in limited scoping studies. Across four sites Jackson et al. [66] found a decreasing trend in bait lamina utilisation with increasing ambient dose rate (gamma air kerma ranged from 0.1–0.5 μGy h$^{-1}$ to 60–138 μGy h$^{-1}$), the lowest feeding activity being observed in a site referred to as the Red Forest (the Jackson et al. study was conducted in 2002). Conversely, unreported data by some of the authors of this paper shows no relationship between feeding activity and soil $^{137}$Cs (range 3–140 kBq kg$^{-1}$ DM) or $^{90}$Sr (range 2–150 kBq kg$^{-1}$ DM) activity concentrations across four CEZ sites (including two towards the western end of the Red Forest) in summer 2005. As these data are unpublished, we have included them within the dataset accompanying this paper [32] to enable independent consideration.

We acknowledge that the endpoint of bait lamina is a measure of invertebrate feeding activity whereas litter bags, used by both Mousseau et al. [19]) and Bonzom et al. [20], give an estimation of organic matter decomposition [67] and consequently the two methods likely predominantly study different organisms. Comparisons between the two approaches differ, with some authors observing similar trends in the results of the two approaches [67–69] and others finding different trends [70, 71]. However, Mousseau et al. [19] imply that their results demonstrated a similar impact of radiation on decomposition by microbial communities and

by detritivorous invertebrates (assessed by comparing the results from fine and coarse mesh litter bags respectively). In future studies, it is important that authors do not extrapolate their findings beyond the limits of the method adopted. A more holistic evaluation of the influence of radiation on soil surface and sub-surface biological activity would be obtained by combining litter decomposition studies with bait lamina deployment.

Rather than current chronic radiation dose rates, soil biological activity varies with soil properties (organic matter content (inferred from soil bulk density), moisture content and pH). These are all well known to influence soil biological activity and feeding activity as determined using bait lamina strips (e.g. [12, 27]). Our finding is in contrast to that of Mousseau et al. [19] who state that their litter bag decomposition data showed a linear dose response of decomposition independent of confounding variables such as pH and soil moisture. The analyses of Mousseau et al. did not appear to consider soil organic matter content as a variable. However, they state that they observed an accumulation of litter with increasing radiation. This is in disagreement with our findings that, irrespective of dose rate, soil biological activity likely increased with increasing soil organic matter content (as inferred for soil bulk density measurements). Furthermore, as noted above, our sites in the Red Forest visually had a sparse litter layer and the most contaminated sites of Mousseau et al. must also have been in the Red Forest (based upon their sample site map). Although our litter layer observation is anecdotal, and cannot be verified, it is supported by our soil bulk density measurements (Table 4).

Whilst we found no relationship between soil invertebrate feeding activity and absorbed dose rate, this does not necessarily mean that radiation has had no impact on soil biota within the CEZ. When analysed by simple habitat category, the Red Forest showed significantly lower feeding activity than the deciduous or coniferous plots. As discussed in Beresford et al. [72] the Red Forest is a unique habitat which was altered by radiation in 1986 and which, at the time of the work reported in this paper, continued to have a relatively poor habitat status. In a review of the impacts of the 1957 Kyshtym (Russian Urals) accident, Fesenko [73] reports that soil invertebrate communities had not been restored at a contaminated site (the main contaminant being $^{90}$Sr) *c.* 30 years after the accident. Fesenko suggested that, in part, continued impacts on soil invertebrates was due to their low mobility and hence a lack of migration into the area. As soil invertebrates have low dispersal rates (e.g. of the order of 5–10 m a$^{-1}$ for earthworms [74]) such a long-term impact of an acute radiation event would seem plausible. Krivolutzkii & Pokarzhevskii [16] report that young earthworms did not survive or hatch in autumn of 1986 close to the ChNPP due to their greater radiosensitivity compared to adults. Therefore, it is possible that the lower biological activity observed in the Red Forest is a residual consequence of what was in effect an acute high exposure to radiation in 1986.

We have estimated total absorbed dose rates for relevant organism types rather than simply using ambient dose rate as a marker of comparative radiation levels across our study sites. In agreement with previous observations [44, 75, 76] we demonstrated that ambient dose rate underestimated total absorbed dose rates to organisms. We recommend that, when relating observations to radiation exposure, total absorbed dose rates are used. This is not a new suggestion (e.g. see Chesser & Baker [77]) but unfortunately bad practice often seems to persist.

There are two aspects of our paper which we would like to draw attention to and encourage as good practice. Firstly, the bait lamina sticks were read 'blind' by people with no knowledge of where they had been deployed (again 'blind' analysis has been recommended previously for studies in the CEZ by Chesser & Baker [77]). Where possible (and it is obviously not possible for observations/measurements made by researchers in the field) blind analysis should be used in future studies as it reduces the potential for bias, either unintentional or intentional, and minimises future criticism. Secondly, there is considerable debate in the scientific literature about the long-term impacts of chronic exposure of wildlife to radiation in the Chernobyl

Exclusion Zone and now also in the Fukushima impacted areas (see Beresford et al. [78]). This lack of consensus has a relatively high public profile, with potential impacts on, for instance, the use of radiation (from medicine to nuclear power) and strategies for remediating areas contaminated by nuclear accidents. By publishing the complete underlying data for our paper [32], we give other scientists the ability to confirm our conclusions or indeed refute them should that be valid. Such an approach of open data publication is the norm in some scientific areas (e.g. for sequencing data, https://www.ncbi.nlm.nih.gov/sra). A wider willingness to make radioecological data freely available in this way would greatly aid the scientific community reaching much needed consensus on the effects of radiation on wildlife under field conditions.

## Supporting information

**S1 Fig. Example profile for the soil CEZ—Plot 18.1 (see Barnett et al. [32]) showing a profile typical for much of the CEZ (including the Red Forest) with little visible organic matter.**
(TIF)

**S2 Fig. Example profile for the soil CEZ—Plot 2.1 (see Barnett et al. [32]) soil profile for site in the deciduous woodland to the south of the Red Forest showing a defined organic matter layer.**
(TIF)

## Acknowledgments

The authors would like to thank Claire Wells (UKCEH) for reading the bait lamina strips, Peter Henrys (UKCEH) for statistical advice, Jacky Chaplow (UKCEH) for help in preparing the accompanying published dataset, Andrey Maksimenko (Chornobyl Center) for radioanalyses and Eugene Guliaichenko (Chornobyl Center) for assistance during sample preparation and fieldwork.

## Author Contributions

**Conceptualization:** Nicholas A. Beresford, Michael D. Wood.

**Data curation:** Catherine L. Barnett.

**Formal analysis:** Nicholas A. Beresford.

**Funding acquisition:** Nicholas A. Beresford, Michael D. Wood.

**Investigation:** Nicholas A. Beresford, Michael D. Wood, Sergey Gashchak, Catherine L. Barnett.

**Methodology:** Nicholas A. Beresford, Michael D. Wood, Sergey Gashchak, Catherine L. Barnett.

**Project administration:** Nicholas A. Beresford.

**Supervision:** Nicholas A. Beresford.

**Writing – original draft:** Nicholas A. Beresford.

**Writing – review & editing:** Nicholas A. Beresford, Michael D. Wood, Sergey Gashchak, Catherine L. Barnett.

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
