## [Decision Letter · Decision Letter 0]

1 Nov 2021

PONE-D-21-24695

Current ionising radiation doses in the Chernobyl Exclusion Zone do not directly impact on soil biological activity

PLOS ONE

Dear Dr. Beresford,

Thank you for submitting your manuscript to PLOS ONE. After careful consideration, we feel that it has merit but does not fully meet PLOS ONE’s publication criteria as it currently stands. Therefore, we invite you to submit a revised version of the manuscript that addresses the points raised during the review process.

We look forward to receiving your revised manuscript.

Kind regards,

Nathalie A. Wall, Dr.

Academic Editor

PLOS ONE

Journal Requirements:

2. Please note that in order to use the direct billing option the corresponding author must be affiliated with the chosen institute. Please either amend your manuscript to change the affiliation or corresponding author, or email us at plosone@plos.org with a request to remove this option.

Reviewers' comments:

Reviewer's Responses to Questions

**Comments to the Author**

1. Is the manuscript technically sound, and do the data support the conclusions?

Reviewer #1: Yes

Reviewer #2: Yes

Reviewer #3: Yes

2. Has the statistical analysis been performed appropriately and rigorously? 

Reviewer #1: Yes

Reviewer #2: Yes

Reviewer #3: Yes

3. Have the authors made all data underlying the findings in their manuscript fully available?

Reviewer #1: Yes

Reviewer #2: Yes

Reviewer #3: Yes

4. Is the manuscript presented in an intelligible fashion and written in standard English?

Reviewer #1: Yes

Reviewer #2: Yes

Reviewer #3: Yes

5. Review Comments to the Author

Reviewer #1: The authors undertook a radiobiological study of the soil fauna in the Chernobyl Exclusion Zone. At many sites with different radiation levels they collected radioecological data and interpreted them in terms of external and internal dose rates of three species of soil organisms. At the same sites, bait lamina soil organism biological activity detectors were installed and exposed for 18 days. Comparison of dose and biological indicators using a variety of statistical methods revealed no effect of radiation. Since the biological effect under study depends not only on the radiation level but also on a number of environmental factors, the detection of the effect of relatively small doses of radiation is a non-trivial task.

The article is of considerable interest to a wide PONE audience interested both in scientific aspects of radiobiology and in practical issues of radiation protection of biota. The experimental field study is carefully performed, and the methods of analysis are adequate to the task at hand. The article is well written and can be published in PONE after the authors have taken into account the reviewer's comments.

Page 5, lines 109-111. For dead trees outside the Red Forest, is there evidence that this death was caused by radiation shortly after the Chernobyl accident?

Pages 7-8. For instruments used to measure dose rates and to analyze soil samples, references to their descriptions or Web sites should be provided.

Page 8, line 186. The assumption that the absorption of the 32-37 keV emission was similar to that of U-isotope emissions should be proved either by measurements or by calculation. In this energy range (13-23 keV and 32-37 keV) the photon absorption strongly depends on the energy.

Page 9, line 205. Which kind of weighting was applied for estimating absorbed dose rates in this study?

Page 9, line 220. Was internal dose rate from alpha emitters multiplied by any radiation weighted factor? That should be explicitly mentioned.

Page 12, Table 2. For better understanding of dose rate pattern in this study it would be useful to present relative contribution of external and internal dose rates (or their ratio), at least for mean values.

Page 15, line 328 and below. Median feeding rates are discussed that are not presented in the paper. I suggest to include them in Table 3 as a separate column.

Page 17, Table 4. In the table heading or as footnote the superscripted letters a and be should be defined.

Page 18, line 403-404. The first sentence of Conclusions reads as ”Our study indicates no effect of current radiation exposure on soil biological activity as determined by bait lamina strips.” In reviewer’s view that conclusion might be too strong taking into account multi-factorial feature of the considered biological endpoint. Authors might consider the following formulation as more appropriate: “Our study did not find any effect of current radiation exposure on soil biological activity as determined by bait lamina strips.”

Reviewer #2: The study is of high quality. Methods and results are clearly stated; results are adequately interpreted and discussed. Conclusions are supported by data. I have no major comments, and I recommend acceptance of the paper.

I found a few typos - please correct:

Line 72: Mousseau et al.. (delete one period)

Line 409: Mousseau et al.. (delete one period)

Line 490: Andrea - should be Andrey

I also have a question about sites 17 and 18 in Buriakivka: was this area affected by fires in April 2015?

Reviewer #3: In this work, the impact of radiation doses on soil biological activity was evaluated using bait lamina as an indicator of in situ microbial activity in the Chernobyl exclusion zone. The strength of this work is to have taken into account a large area with 53 study plots allowing to explore a wide range of radiation doses. The other strength is the consideration of absorbed dose rather than dose rate to estimate the effects. The different results obtained as well as the comparison with the data, sometimes divergent, of the literature are discussed in a clear, rigorous and convincing way. This paper represents a very welcome contribution to the field of radioecology, especially in understanding the consequences of the Chernobyl accident which is still under debate.

I have no additional comments and only one suggestion:

line 317: please give values for temperatures (in situ and lower bound).

6. PLOS authors have the option to publish the peer review history of their article (what does this mean?). If published, this will include your full peer review and any attached files.

Reviewer #1: **Yes: **Prof. Mikhail Balonov

Reviewer #2: No

Reviewer #3: No

---

## [Author Response · Author response to Decision Letter 0]

9 Dec 2021

We note that Figure 1 in your submission contain [map/satellite] images which may be copyrighted. All PLOS content is published under the Creative Commons Attribution License (CC BY 4.0), which means that the manuscript, images, and Supporting Information files will be freely available online, and any third party is permitted to access, download, copy, distribute, and use these materials in any way, even commercially, with proper attribution. For these reasons, we cannot publish previously copyrighted maps or satellite images created using proprietary data, such as Google software (Google Maps, Street View, and Earth).

RESPONSE>>The figure was produced by Chornobyl Center and has no previous copyright. Dr Gashchak (co-author and Deputy Director of the Chornobyl Center) has signed the Content Permission Form which we have uploaded).

Note we have reviewed formatting requirements etc. and the paper has been amended where required to meet these.

Review Comments to the Author

Reviewer #1: The authors undertook a radiobiological study of the soil fauna in the Chernobyl Exclusion Zone. At many sites with different radiation levels they collected radioecological data and interpreted them in terms of external and internal dose rates of three species of soil organisms. At the same sites, bait lamina soil organism biological activity detectors were installed and exposed for 18 days. Comparison of dose and biological indicators using a variety of statistical methods revealed no effect of radiation. Since the biological effect under study depends not only on the radiation level but also on a number of environmental factors, the detection of the effect of relatively small doses of radiation is a non-trivial task.

The article is of considerable interest to a wide PONE audience interested both in scientific aspects of radiobiology and in practical issues of radiation protection of biota. The experimental field study is carefully performed, and the methods of analysis are adequate to the task at hand. The article is well written and can be published in PONE after the authors have taken into account the reviewer's comments.

RESPONSE>>We thank the reviewer for their positive comments.

Page 5, lines 109-111. For dead trees outside the Red Forest, is there evidence that this death was caused by radiation shortly after the Chernobyl accident?

RESPONSE>>We have amended the wording as suggested by the reviewer on the marked manuscript and also put a note into the text that dose rates at these sites were higher that at some Red Forest sites.

Pages 7-8. For instruments used to measure dose rates and to analyze soil samples, references to their descriptions or Web sites should be provided.

RESPONSE>> Supplier details have been added for all instruments etc..

Page 8, line 186. The assumption that the absorption of the 32-37 keV emission was similar to that of U-isotope emissions should be proved either by measurements or by calculation. In this energy range (13-23 keV and 32-37 keV) the photon absorption strongly depends on the energy.

RESPONSE>> The text has been amended to clarify this aspect of the methods for readers.

Page 9, line 205. Which kind of weighting was applied for estimating absorbed dose rates in this study?

RESPONSE>> The ERICA Tool default radiation weighting factors of 10 for alpha radiation, 3 for low energy beta and 1 for high energy beta and gamma radiation were used. We have now clarified this in the text.

Page 9, line 220. Was internal dose rate from alpha emitters multiplied by any radiation weighted factor? That should be explicitly mentioned.

RESPONSE>>As noted above we have clarified how weighted dose rates were calculated.

Page 12, Table 2. For better understanding of dose rate pattern in this study it would be useful to present relative contribution of external and internal dose rates (or their ratio), at least for mean values.

RESPONSE>>The percentage contributions of internal exposure are now presented in text.

Page 15, line 328 and below. Median feeding rates are discussed that are not presented in the paper. I suggest to include them in Table 3 as a separate column.

RESPONSE>>Median values have been added in text.

Page 17, Table 4. In the table heading or as footnote the superscripted letters a and be should be defined.

RESPONSE>>These are explained in the Table title: For a given parameter significant differences (p<0.05; generalised linear model) between habitats are identified by different superscripted letter.

Page 18, line 403-404. The first sentence of Conclusions reads as ”Our study indicates no effect of current radiation exposure on soil biological activity as determined by bait lamina strips.” In reviewer’s view that conclusion might be too strong taking into account multi-factorial feature of the considered biological endpoint. Authors might consider the following formulation as more appropriate: “Our study did not find any effect of current radiation exposure on soil biological activity as determined by bait lamina strips.”

RESPONSE>>Amended as suggested.

NOTE – we have reviewed and additional comments on the marked manuscript from this reviewer and edited our paper accordingly. 

Reviewer #2: The study is of high quality. Methods and results are clearly stated; results are adequately interpreted and discussed. Conclusions are supported by data. I have no major comments, and I recommend acceptance of the paper.

RESPONSE>>We thank the reviewer for their positive comments.

I found a few typos - please correct:

Line 72: Mousseau et al.. (delete one period)

Line 409: Mousseau et al.. (delete one period)

RESPONSE>>As these appear at the end of a sentence they should have two periods.

Line 490: Andrea - should be Andrey

RESPONSE>> Corrected.

I also have a question about sites 17 and 18 in Buriakivka: was this area affected by fires in April 2015?

RESPONSE>>These sites had not been impact by wildfires. We have clarified that no study site had been impacted by wildfires in the text.

Reviewer #3: In this work, the impact of radiation doses on soil biological activity was evaluated using bait lamina as an indicator of in situ microbial activity in the Chernobyl exclusion zone. The strength of this work is to have taken into account a large area with 53 study plots allowing to explore a wide range of radiation doses. The other strength is the consideration of absorbed dose rather than dose rate to estimate the effects. The different results obtained as well as the comparison with the data, sometimes divergent, of the literature are discussed in a clear, rigorous and convincing way. This paper represents a very welcome contribution to the field of radioecology, especially in understanding the consequences of the Chernobyl accident which is still under debate.

RESPONSE>>We thank the reviewer for their very positive comments.

I have no additional comments and only one suggestion:

line 317: please give values for temperatures (in situ and lower bound).

RESPONSE>>Soil temperatures are presented in Table 4.

---

## [Editor Report · Decision Letter 1]

24 Jan 2022

Current ionising radiation doses in the Chernobyl Exclusion Zone do not directly impact on soil biological activity

PONE-D-21-24695R1

Dear Dr. Beresford,

We’re pleased to inform you that your manuscript has been judged scientifically suitable for publication and will be formally accepted for publication once it meets all outstanding technical requirements.

Kind regards,

Nathalie A. Wall, Dr.

Academic Editor

PLOS ONE
---

## [Editor Report · Acceptance letter]

28 Jan 2022

PONE-D-21-24695R1 

Current ionising radiation doses in the Chernobyl Exclusion Zone do not directly impact on soil biological activity 

Dear Dr. Beresford:

I'm pleased to inform you that your manuscript has been deemed suitable for publication in PLOS ONE. Congratulations! Your manuscript is now with our production department. 

Kind regards, 

on behalf of

Prof. Nathalie A. Wall 

Academic Editor

PLOS ONE